# Kisspeptin Is Upregulated at the Maternal-Fetal Interface of the Preeclamptic-like BPH/5 Mouse and Normalized after Synchronization of Sex Steroid Hormones

Viviane C. L. Gomes [1], Ashley K. Woods [2], Kassandra R. Crissman [1], Camille A. Landry [1], Kalie F. Beckers [1], Bryce M. Gilbert [3], Lucas R. Ferro [1], Chin-Chi Liu [1], Erin L. Oberhaus [3] and Jenny L. Sones [1,*]

1 Department of Veterinary Clinical Sciences, Louisiana State University School of Veterinary Medicine, Baton Rouge, LA 70803, USA
2 Department of Biomedical Sciences, Cornell University College of Veterinary Medicine, Ithaca, NY 14853, USA
3 School of Animal Sciences, Louisiana Agricultural Experiment Station, LSU AgCenter, Baton Rouge, LA 70803, USA
* Correspondence: jsones@lsu.edu

**Abstract:** Insufficient invasion of conceptus-derived trophoblast cells in the maternal decidua is a key event in the development of early-onset preeclampsia (PE), a subtype of PE associated with high maternal and fetal morbidity and mortality. Kisspeptins, a family of peptides previously shown to inhibit trophoblast cell invasion, have been implicated in the pathogenesis of early-onset PE. However, a role of kisspeptin signaling during the genesis of this syndrome has not been elucidated. Herein, we used the preeclamptic-like BPH/5 mouse model to investigate kisspeptin expression and potential upstream regulatory mechanisms in a PE-like syndrome. Expression of the kisspeptin encoding gene, *Kiss1*, and the 10-amino-acid kisspeptide (Kp-10), are upregulated in the non-pregnant uterus of BPH/5 females during diestrus and in the maternal-fetal interface during embryonic implantation and decidualization. Correspondingly, the dysregulation of molecular pathways downstream to kisspeptins also occurs in this mouse model. BPH/5 females have abnormal sex steroid hormone profiles during early gestation. In this study, the normalization of circulating concentrations of 17β-estradiol (E2) and progesterone (P4) in pregnant BPH/5 females not only mitigated *Kiss1* upregulation, but also rescued the expression of multiple molecules downstream to kisspeptin and ameliorated adverse fetoplacental outcomes. Those findings suggest that uterine *Kiss1* upregulation occurs pre-pregnancy and persists during early gestation in a PE-like mouse model. Moreover, this study highlights the role of sex steroid hormones in uteroplacental *Kiss1* dysregulation and the improvement of placentation by normalization of E2, P4 and *Kiss1*.

**Keywords:** *Kiss1/Kiss1r*; *GPR54*; trophoblast; estrogen; progesterone

## 1. Introduction

Preeclampsia (PE) is a hypertensive disorder of pregnancy that contributes significantly to maternal and fetal morbidity and mortality. Globally, the PE-eclampsia complex is estimated to contribute to approximately 63,000 maternal deaths per year [1]. PE is a multiorgan disorder, mainly characterized by new-onset hypertension after 20 weeks of gestation (systolic blood pressure > 140 mmHg and diastolic blood pressure > 90 mmHg), glomerular endotheliosis, proteinuria ($\geq$300 mg of protein/day or protein/creatinine ratio $\geq$ 30 mg/mmol) or another accompanying sign/symptom of organ dysfunction [2]. Based on the multifactorial etiopathogenesis and the array of clinical presentations, at least two subtypes of PE are recognized [3]. These are early and late-onset PE, characterized by the development of clinical signs before or after 34–35 weeks of gestation, respectively [4,5]. Early-onset PE often leads to more severe maternal and fetal adverse outcomes, including fetal growth restriction, low birth weight and infant perinatal death [4].

Inadequate trophoblast cell invasion and poor remodeling of the uterine spiral arteries are recognized as key events in the pathogenesis of early-onset PE [6,7]. During early gestation, the extravillous cytotrophoblast cells (evCT) detach from the placenta anchoring villi, migrate interstitially through the maternal decidua and inner third of the myometrium and endovascularly towards the lumen of the uterine spiral arteries, promoting remodeling and decreased vascular resistance [6–10]. Numerous mechanisms modulate evCT invasion, including a balance between extracellular matrix degradation by trophoblast-secreted matrix metalloproteinases (MMPs) and endogenous MMP inhibition via non-covalent binding with tissue inhibitors of metalloproteinases (TIMPs) [9,11,12]. The incomplete understanding of spatial and temporal regulation of evCT dynamics continues to hinder a holistic comprehension of PE etiopathogenesis and, potentially, the development of diagnostic tools and targeted interventions during the early development of this syndrome.

Kisspeptins are a family of small peptides containing 10 to 54 amino acids (KP-10, KP-13, KP-14 and KP-54), derived from the post-translational cleavage of the *KISS1* gene product (KP-145). Kisspeptins may exert important roles as physiological modulators of decidualization, embryonic implantation, and evCT invasion [13]. The kisspeptin-mediated inhibition of trophoblast cell invasion has been demonstrated in vitro, and treatment of first trimester human trophoblast cells with kisspeptin leads to the upregulation of TIMPs and downregulation of MMPs [14–17]. The expression of kisspeptins and their cognate G protein-coupled receptor (*KISS1R*) are higher in human non-pregnant decidualized endometrium, and in healthy placentas during the peak of trophoblast cell invasion [14,16,18]. Additionally, low *KISS1/KISS1R* expression was described in the highly invasive trophoblast cells of choriocarcinoma, while high *KISS1/KISS1R* expression was reported in the non-invasive cells of benign molar pregnancies [16].

Kisspeptin expression is higher in term placentas of women with early-onset PE than in those with uncomplicated pregnancies [19–21]. Nonetheless, the expression of kisspeptins during the genesis of early-onset PE has not been investigated, and the mechanisms leading to kisspeptin upregulation at the maternal-fetal interface remain largely unexplored. A regulatory interaction between sex steroid hormones (SSH) and KISS1/KISS1R is speculated. In non-pregnant women, uterine KISS1/KISS1R vary according to the SSH milieu of the menstrual cycle [18]. Moreover, increased uterine and placental *Kiss1* and *Kiss1r* expression have been demonstrated in mice after the administration of 17β-estradiol (E2) and/or progesterone (P4), and in human trophoblast cells in vitro [22,23].

The Blood Pressure High Subline 5 (BPH/5) is a mouse model that spontaneously develops the main features of PE, including late gestational hypertension, endothelial dysfunction and proteinuria [24]. Comparable to the human syndrome, the clinical signs subside after delivery of fetuses and placentas [24]. BPH/5 females present abnormal embryonic implantation, defective placentation, decreased trophoblast cell invasion, reduced placental labyrinth endothelial remodeling, and fetal growth restriction [24–27]. Interestingly, BPH/5 females present aberrant estrous cycles compared to control C57BL/6 (C57) females, and abnormal circulating SSH concentrations pre-pregnancy and during early gestation [27,28].

Herein, the BPH/5 mouse was used to study the expression, regulatory mechanisms, and function of kisspeptins in a PE-like syndrome. It was hypothesized that, secondary to an abnormal SSH profile, kisspeptin and the kisspeptin receptor would be upregulated in the BPH/5 non-pregnant uterus during diestrus and at the maternal-fetal interface during early gestation. Additionally, it was hypothesized that abnormal uteroplacental kisspeptin signaling would be associated with the defective implantation and placentation reported in the BPH/5 mouse model. The interaction between SSH and uteroplacental kisspeptin signaling was investigated by assessing the effects of early gestation artificial synchronization (AS-SSH) of E2 and P4 on *Kiss1* and downstream molecular expression and concurrent placental development. To date, this is the first known report to investigate the potential roles and regulation of kisspeptins using an animal model of PE.

## 2. Materials and Methods

### 2.1. Animal Husbandry

Experiments were performed using virgin BPH/5 and control C57 females from in-house colonies. The C57 is a normotensive mouse strain originally used in the eight-way cross that led to the development of BPH/5 [24]. Adult females (8–12 weeks of age) were housed in a climate-controlled environment (12-h light-dark cycle, 70.5–71 °F) and fed a standard chow diet (Purina 5001) and ad libitum water. All animal procedures were approved by the Institutional Animal Care and Use Committees at Cornell University, Louisiana State University School of Veterinary Medicine, or Pennington Biomedical Research Center and are in accordance with the PHS Guide for the Care and Use of Laboratory Animals.

### 2.2. Reproductive Management and Sample Collection

Virgin BPH/5 and C57 females singly housed were used for estrous cycle staging and sample collection during nonpregnant diestrus (NP-D; $n = 3$–7/group). Vaginal cytology samples were collected daily from nonpregnant BPH/5 and C57 females for at least two complete estrous cycles, in accordance with previous reports [29,30]. Nonpregnant females were euthanized during the first day of cytological diestrus for sample collection (NP-D). Additionally, strain-matched timed matings were performed, with the day of detection of a vaginal copulatory plug designated as embryonic (e) 0.5. Single-housed BPH/5 and C57 females either carried natural (NAT) pregnancies or were randomly assigned to AS-SSH during early gestation (Figure 1). NAT BPH/5 and NAT C57 were euthanized at e4.5 and e7.5 ($n = 5$–9/group), and embryonic implantation sites were harvested for gene and protein expression studies. For investigation of gene expression during the peak of embryonic implantation in AS-SSH pregnancies, AS-SSH BPH/5 and AS-SSH C57 were euthanized two days post-E2 injection ($n = 8$/group). Approximately 10 min before euthanasia and sample collection, a single injection of 100 µL of 1% Evan's Blue dissolved in saline was administered via tail vein injection in pregnant mice to aid in gross identification of early gestation embryonic implantation sites, as previously described [27]. Additional NAT and AS-SSH BPH/5 and C57 females were anesthetized after the placenta was formed, at e12.5 in NAT pregnancies and 9 days post-E2 injection in AS-SSH pregnancies, for ultrasound examination of individual fetoplacental units ($n = 16$–69/group), followed by post-mortem sample collection and placental morphometry ($n = 5$–8/group).

### 2.3. Artificial Synchronization of Sex Steroid Hormones

Early gestation AS of E2 and P4 was performed in BPH/5 and C57 females by adapting the previously described mouse delayed-implantation model [31]. Strain-matched timed matings were performed and pregnant BPH/5 and C57 females underwent ovariectomy in the afternoon of e2.5. The females were anesthetized with ketamine (100 mg/kg) and xylazine (10 mg/kg) administered intraperitoneally, and dorsal incisions were performed to exteriorize and excise the ovaries. A single dose of E2 (25 ng/mouse, subcutaneously) was administered one day after surgery to stimulate embryonic implantation, and daily subcutaneous injections of P4 (1 ng/mouse) were given until the day of sample collection (Figure 1).

### 2.4. Quantitative Reverse-Transcription Polymerase Chain Reaction (qRT-PCR)

Samples from BPH/5 and C57 NP-D uterus and embryonic implantation sites (eIS) from NAT and AS-SSH pregnancies were collected immediately after euthanasia. Aliquots were flash-frozen and cryopreserved at −80 °C until further analysis. Genomic DNA was eliminated, and total RNA was extracted using a commercial kit, according to the manufacturer's instructions (Qiagen RNeasy, Hilden, Germany). The RNA ratio of absorbance and concentration were assessed using a NanoDrop Spectrophotometer (NanoDrop 200, ThermoFisher Scientific, Wilmington, NC, USA) and 1000 ng of cDNA were synthetized using a commercial kit for reverse transcription (qScript cDNA, Quanta Biosciences, Gaithersburg, MD, USA). Quantification of gene expression levels was performed by qRT-PCR using SYBR Green

(PerfeCTa SYBR Green FastMix, Quanta Biosciences, Gaithersburg, MD, USA). Each sample was run in triplicate and mRNA expression was normalized to 18S and analyzed using the ddCT method [32]. Sequence-specific amplification was confirmed by a single peak during the dissociation protocol following amplification and by product size using gel electrophoresis. Gene targets and primer sequences are listed in Supplementary Table S1.

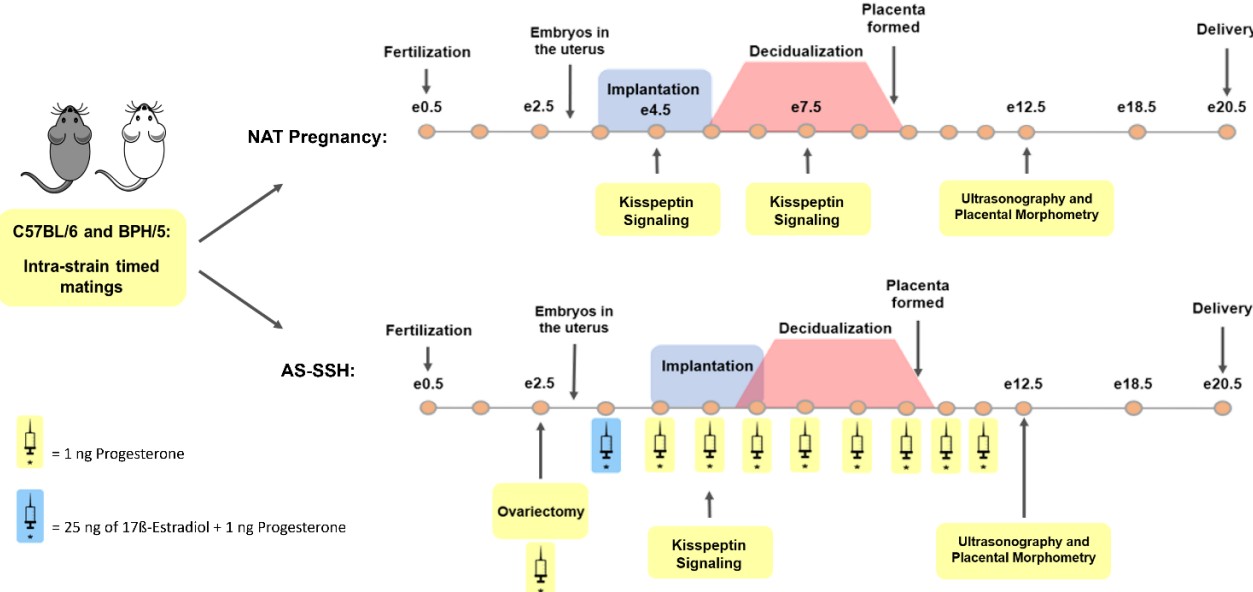

**Figure 1.** Sample collection of natural (NAT) pregnancies and after artificial synchronization of sex steroid hormone profile (AS-SSH) in the preeclamptic-like BPH/5 mouse and C57BL/6 (C57) normotensive controls. In NAT pregnancies, embryonic implantation sites (eIS) of strain matched pregnancies were collected during the peak of embryonic implantation and decidualization at embryonic days (e) 4.5 and e7.5, respectively. For AS-SSH, strain-matched timed matings were performed and pregnant BPH/5 and C57 females underwent ovariectomy in the afternoon of e2.5. A single dose of 17β-estradiol (E2, 25 ng/mouse, subcutaneously) was administered one day after surgery to stimulate embryonic implantation, and daily subcutaneous injections of progesterone (P4, 1ng/mouse) were given until the day of sample collection. Two days post E2 injection, eIS were harvested from AS-SSH BPH/5 and AS-SSH C57 females for investigation of kisspeptin expression and signaling. A cohort of NAT and AS-SSH females was anesthetized at e12.5 and 9 days post-E2 administration, respectively, for ultrasonographic assessment of fetal heart rate and umbilical blood flow. Collection of fetoplacental units was then performed for placental morphometry.

*2.5. Immunohistochemistry*

The uterus and eIS were collected immediately after euthanasia and fixed using 4% paraformaldehyde for 24 h, followed by ethanol 70%, and embedded in paraffin for immunohistochemistry assays (*n* = 3–4/group). Tissue sections (4 μm) were deparaffinized and rehydrated using xylene and a graded series of ethanol solutions, namely, ethanol 100%, 95%, 70% and 50%. Antigen retrieval was performed using sodium citrate at 90–95 °C. Endogenous peroxidase activity was blocked by incubating the slides in 0.3% hydrogen peroxidase in methanol for 30 min at room temperature. Non-specific binding was prevented by incubating tissue sections in normal goat serum for 30 min at room temperature. The tissue sections were then incubated with primary antibody for one hour at room temperature, then overnight at 4 °C. Polyclonal rabbit anti-human KP-10 antibody (1:400, Millipore, Burlington, MA, USA) was utilized based on previous immunohistochemistry studies of mouse uterus and decidua, and validation using mouse hypothalamic samples [22]. A 90% sequence homology occurs between human KP-10 and mouse Kp-10. The primary antibody was omitted in negative control samples. After rinsing in Tris-buffered saline with Triton X, the sections were incubated with secondary biotinylated goat anti-rabbit IgG

for 1 h at room temperature (1:500), followed by incubation in avidin-biotin complex at room temperature for 1 h (Vectastain Elite ABC-HRP, Vector Laboratories, Newark, CA, USA). Diaminobenzidine solution was used to detect immunostaining. Counterstaining was performed with hematoxylin. Quantification of Kp-10 immunostaining was performed using Image J® software (NIH, version 1.33). Specifically, 100 cytoplasmic areas of NP-D uterine stroma (US), glandular epithelium (GE) and luminal epithelium, as well as e7.5 US, GE, inter-implantation site (IIS LE), and decidua (DE) were randomly selected by an observer blinded to the study design. The images were standardized and converted to gray scale, and the mean gray value per pixel was measured as a marker of staining intensity. The experiments were performed in triplicate for each timepoint.

### 2.6. Ultrasonography

For the ultrasonographic assessment of pregnancy, BPH/5 and C57 females from the NAT and AS-SSH groups were anesthetized using 2% isoflurane and perioperative body temperature monitoring and maintenance were performed as previously described [27]. A ventral midline celiotomy was performed, the uterus was exteriorized, and ultrasound examination of individual fetoplacental units was performed using Vevo 770 ultra-high-frequency ultrasound. The fetal heart rates (beats/min) were measured and reported as a ratio of fetal heart rate to maternal heart rate. Umbilical blood flow (mm/sec) was assessed via Doppler ultrasound measurements of umbilical vessels. Euthanasia ensued immediately after ultrasound and fetoplacental units were harvested for placental morphometry.

### 2.7. Placental Morphometry

Fetoplacental units from NAT and AS-SSH BPH/5 and C57 pregnancies were collected, fixed in 10% neutral buffered formalin and embedded in paraffin (*n* = 4–6/group). Sections of fetoplacental units (4 μm) were stained using Isolectin B4, as previously described [25]. Placental decidual expansion was assessed by an observer blinded to the experimental design using Image J® software (NIH, version 1.33). The placental expansion into the maternal decidua was calculated as the ratio of the labyrinth and junctional zone (placenta) in relation to the entire placental disc (placenta + decidua) [25].

### 2.8. Statistical Analysis

Data analyses were performed using GraphPad Prism, version 9.4 (GraphPad Prism Software, Inc., La Jolla, CA, USA). Student's t-tests were used for comparisons between estrous-cycle- or gestational-age-matched BPH/5 and C57. Welch's corrections were performed for inequality of variances. One-way ANOVA was used for comparison of embryonic implantation rate in NAT and AS-SSH BPH/5 and C57 pregnancies. Normality of residuals from the models were accessed and confirmed via Shapiro-Wilk tests and quantile-quantile (Q-Q) plots. Kruskal-Wallis and post-hoc Dunn's tests were used for comparisons between multiple groups if the residuals from the models were not normally distributed. Data are presented as mean ± SEM. Significance was set at $p < 0.05$.

## 3. Results

### 3.1. Kiss1 Is Upregulated in BPH/5 Non-Pregnant Uterus and Maternal-Fetal Interface

In non-pregnant women, kisspeptin expression is higher in the decidualized endometrial stromal cells during the late secretory phase of the menstrual cycle when compared to proliferative or early secretory phases and post-menopause [18]. Therefore, a role of kisspeptin signaling during uterine preparation for embryonic implantation has been proposed. We sought to characterize uterine kisspeptin/receptor expression in BPH/5 females in the non-pregnant luteal phase, as the pre-pregnancy uterine molecular profile is vastly unexplored in the context of PE. During NP-D, *Kiss1* is approximately fourfold higher in BPH/5 females compared to C57 (Figure 2A; *p* = 0.049), while uterine *Kiss1r* expression is not different between NP-D BPH/5 and NP-D C57 (Supplementary Figure S1A; *p* > 0.05). During NP-D, Kp-10 immunostaining is mainly located in the uterine LE and GE of BPH/5

and C57 females, with significantly higher staining intensity when compared to strain matched US (Figure 2B–H; *p* < 0.001). While Kp-10 immunostaining intensity is not different between the US of BPH/5 and C57 females during NP-D, it is higher in the uterine GE and LE of BPH/5 females in comparison to C57 controls (Figure 2B–H; *p* < 0.001).

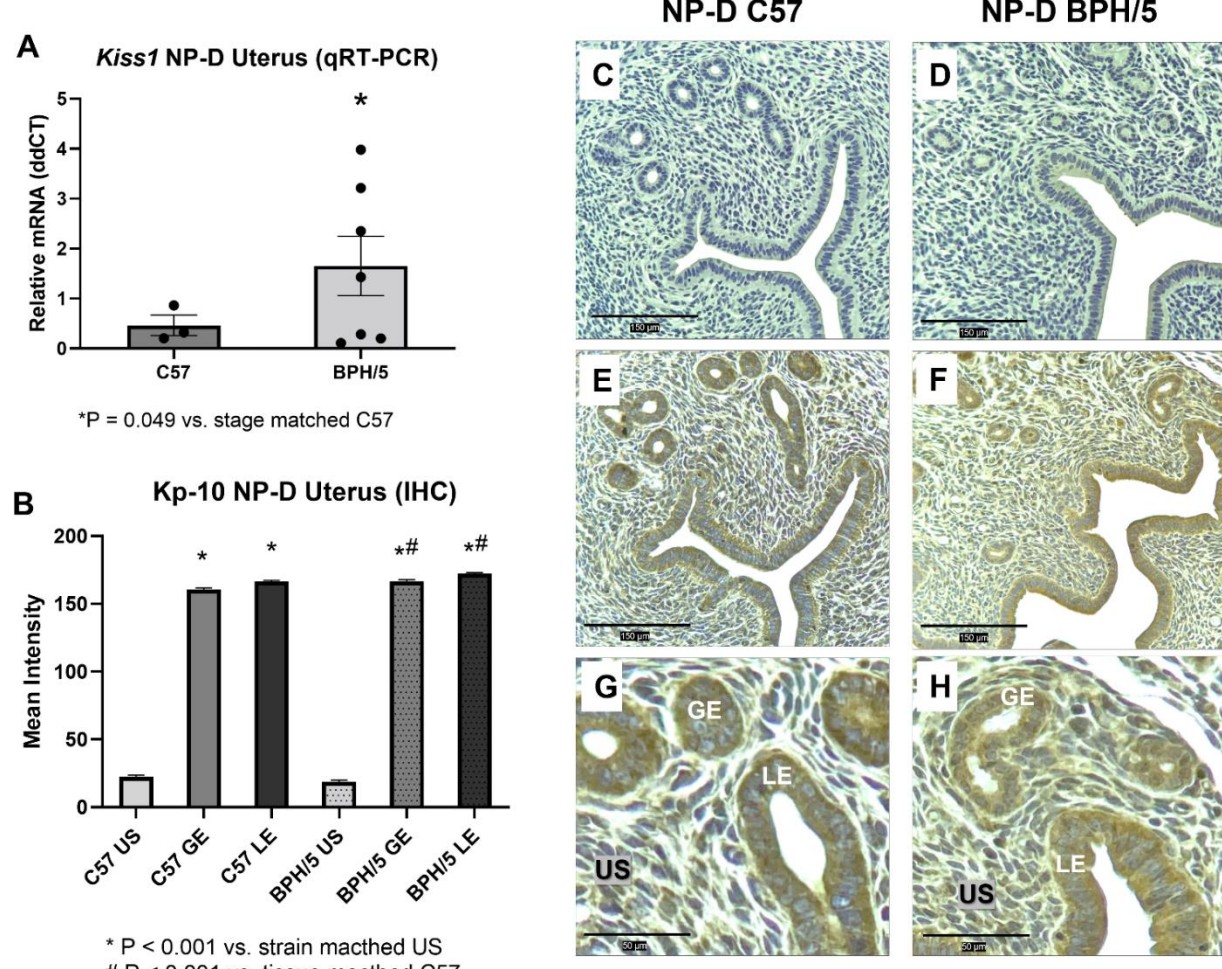

**Figure 2.** Kisspeptins are upregulated in the BPH/5 non-pregnant uterus during diestrus. (**A**) Quantification of *Kiss1* gene expression in the non-pregnant diestrus (NP-D) uterus via qRT-PCR (*n* = 3–7 females/group). Student's *t*-test, * *p* = 0.049 vs. estrous cycle stage matched C57. (**B**) Quantification of mean cytoplasmic immunohistochemical staining intensity of kisspeptin-10 (Kp-10) in the uterine stroma (US), glandular epithelium (GE) and luminal epithelium (LE) of BPH/5 and C57 females during NP-D (*n* = 3/group). Kruskal-Wallis and post-hoc Dunn's test, * *p* < 0.001 vs. strain matched US, # *p* < 0.001 vs. tissue matched of NP-D C57. Data expressed as mean ± SEM. (**C–H**) Representative images of Kiss1 immunostaining in BPH/5 and C57 NP-D uteri: (**C,D**) Negative control, scale bar = 150 μm; (**E,F**) Kiss1, scale bar = 150 μm; (**G,H**) Kiss1, scale bar = 50 μm.

In healthy mouse pregnancies, blastocysts enter the uterus at approximately 00:00 on e3.5, and implantation occurs between the evening of e3.5 and e5.5 [23,33]. Delayed embryonic implantation has been previously described in the BPH/5 mouse. While signs of embryonic implantation are first seen in C57 females in the evening of e3.5, embryonic implantation sites (eIS) are only noted in BPH/5 females in the morning of e4.5 [27]. Interestingly, *Kiss1* is approximately fourfold higher in NAT BPH/5 vs. NAT C57 eIS at e4.5 (Figure 3A; *p* = 0.013), while eIS *Kiss1r* is not different between NAT BPH/5 and NAT C57 females (Supplementary Figure S1F; *p* > 0.05). Steroid hormone-induced decidualization ensues after embryonic attachment in mice, with extensive proliferation and

differentiation of uterine stromal cells between e4.5 and e7.5 [34]. Following abnormal embryonic implantation, BPH/5 females also present delayed decidualization, with an apparent overexpression of decidual markers beginning in the evening of e4.5 [27]. During the peak of decidualization, at e7.5, *Kiss1* and *Kiss1r* are approximately twofold higher in NAT BPH/5 when compared to NAT C57 eIS (Figure 3F; $p = 0.044$, and Supplementary Figure S1C; $p = 0.023$, respectively). At e7.5, Kp-10 immunostaining was mainly identified in the GE and IIS LE, but also in the decidualized stromal cells (DE) of NAT BPH/5 and C57 eIS (Figure 3C–O). Interestingly, Kp-10 immunostaining intensity was higher in NAT BPH/5 GE and DE when compared to the same tissue in NAT C57 (Figure 3C,G–I and M–O; $p < 0.001$), whereas no difference in staining intensity was noted in the IIS LE of NAT BPH/5 and C57 females (Figure 3C,J–L; $p > 0.05$).

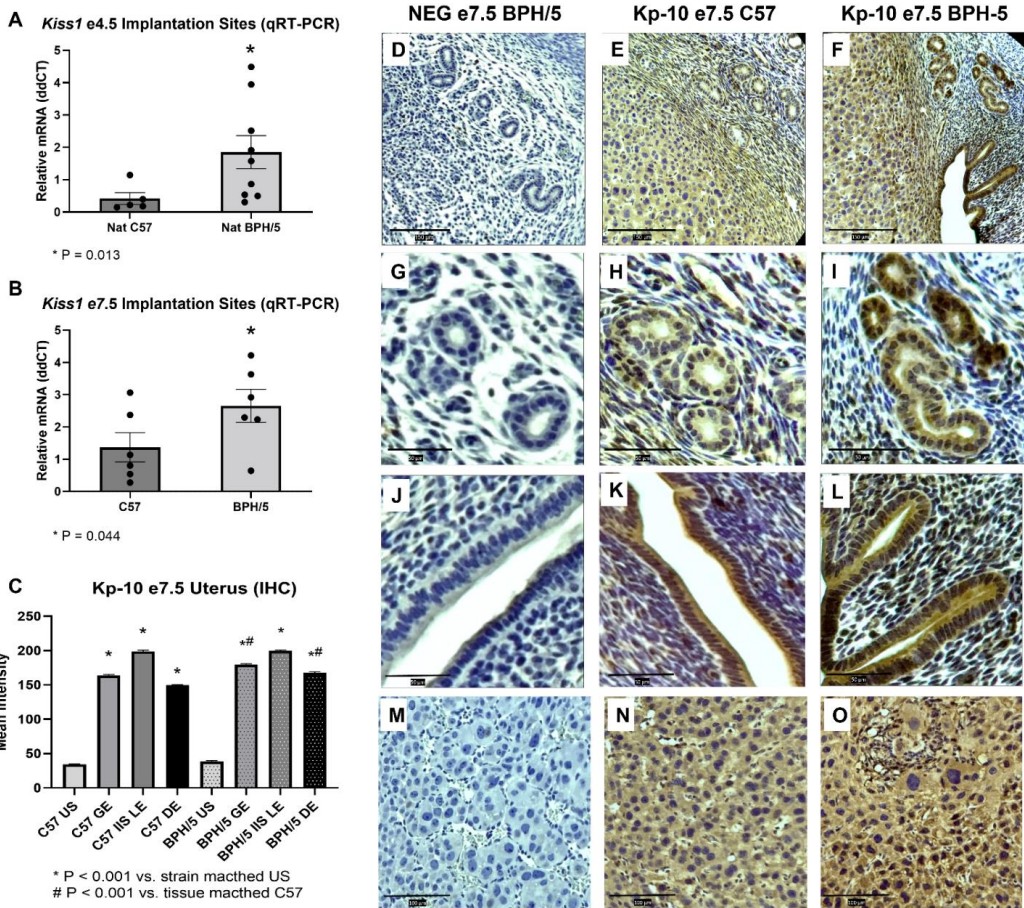

**Figure 3.** Kisspeptin is upregulated in the BPH/5 embryonic implantation sites (eIS) during the peak of implantation and decidualization. (**A**,**B**) Quantification of *Kiss1* gene expression in the eIS of natural (NAT) BPH/5 and NAT C57 pregnancies via qRT-PCR. (A) *Kiss1* expression is higher in the eIS of NAT BPH/5 vs. NAT C57 during embryonic implantation, at embryonic day (e) 4.5 ($n = 5$–9/group). Student's *t*-test, * $p = 0.013$. (**B**) *Kiss1* expression is higher in the eIS of NAT BPH/5 vs. NAT C57 during the peak of decidualization, at e7.5 ($n = 6$/group). Student's *t*-test, and * $p = 0.044$. (**C**) Quantification of mean cytoplasmic kisspeptin-10 (Kp-10) staining intensity in the uterine stroma (US), glandular epithelium (GE), inter-implantation site luminal epithelium (IIS LE) and decidualized stromal cells (DE) of NAT BPH/5 and C57 pregnancies via immunohistochemistry ($n = 3$/group). Kruskal-Wallis and post-hoc Dunn's test, * $p < 0.001$ vs. strain matched US, # $p < 0.001$ vs. tissue matched NAT C57. Data expressed as mean ± SEM. (**D–O**) Representative images of negative control (**D,G,J,M**) and Kp-10 immunostaining in NAT C57 (**E,H,K,N**) and NAT BPH/5 (**F,I,L,O**) uterus and placenta at e7.5. (**D–F**) scale bar = 150 μm; (**G–L**) scale bar = 50 μm; (**M–O**) scale bar = 100 μm.

### 3.2. Timps Are Upregulated in the BPH/5 Non-Pregnant Uterus and Maternal-Fetal Interface

To further investigate the kisspeptin modulation of evCT invasion, the uteroplacental expression of *Timps* was studied in BPH/5 females via qRT-PCR (Figure 4). Four TIMPs have been described in vertebrates, namely TIMP-1, -2, -3 and -4, and important roles of those molecules in hypertension and PE have been suggested [35]. Importantly, kisspeptins have been previously shown to induce upregulation of *TIMPs* in first-trimester human trophoblast cells [17]. Correspondingly, uteroplacental upregulation of *Timps* in BPH/5 were seen in the time points investigated. In NP-D, *Timp1* and *Timp2* are approximately nine- and 14-fold higher in NAT BPH/5 females compared to NAT C57 (Figure 4A,D; $p = 0.035$ and $p = 0.015$, respectively). During embryonic implantation (e4.5), eIS expression of *Timp1*, *Timp2* and *Timp4* are 3-, 7-, and 6-fold higher, respectively, in NAT BPH/5 when compared to NAT C57 (Figure 4B,E,H; $p = 0.04$, $p = 0.004$ and $p = 0.019$, respectively). Furthermore, *Timp1* and *Timp2* are approximately 2- and 13-fold higher, respectively, in NAT BPH/5 eIS vs. NAT C57 during the peak of decidualization (Figure 4C,F; $p = 0.044$ and $p = 0.047$, respectively). While *Timp4* is upregulated in NAT BPH/5 eIS at e4.5, it is not different between BPH/5 and C57 during NP-D and NAT pregnancies at e7.5 (Figure 4G,I; $p > 0.05$).

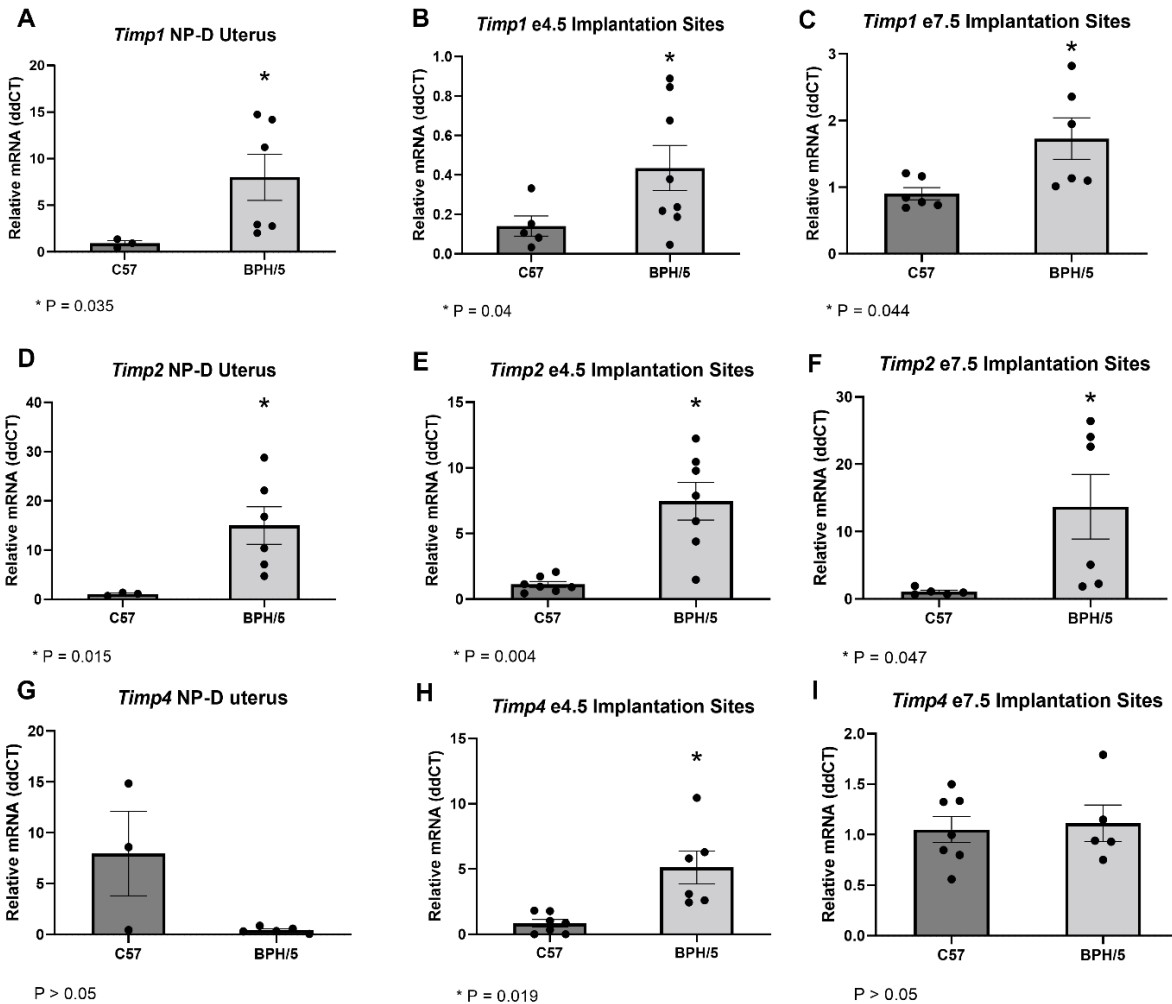

**Figure 4.** *Timps* are upregulated in the BPH/5 non-pregnant uterus and maternal-fetal interface. Quantification of *Timp-1* (**A–C**), *-2* (**D–F**) and *-4* (**G–I**) gene expression in the non-pregnant diestrus (NP-D) uterus and embryonic implantation sites (eIS) of natural (NAT) BPH/5 females and C57 females via qRT-PCR ($n = 3–8$/group). Student's *t*-test, * $p < 0.05$ vs. time-matched C57. Data expressed as mean ± SEM.

### 3.3. Artificial Synchronization of SSH Normalizes the Expression of Kiss1 and Downstream Molecules in the BPH/5 Mouse

Kisspeptin/receptor upregulation in mouse uterus and placenta, as well as in human trophoblast cells, have been demonstrated after administration of E2 and/or P4 [22,23]. In NAT BPH/5 pregnancies, a premature and depressed E2 surge occurs in the morning of e2.5, and higher circulating P4 concentrations are seen in the morning and evening of e2.5 [27,28]. This abnormal SSH profile may contribute to implantation defects seen in this model [27]. In NAT C57 pregnancies, E2 surge occurs in the afternoon of e2.5 and E2 concentrations are significantly higher than in NAT BPH/5 females until the morning of e3.5 [27]. In NAT C57 females, increased vascular permeability characteristic of embryonic implantation is evident after macromolecular blue dye staining in the morning of e4.5 [27,33]. In this study, the expression of *Kiss1/Kiss1r* and the proposed downstream molecules *Timp1*, *Timp2* and *Timp4* was investigated during the window of embryonic implantation (2 days post-E2 administration) in BPH/5 and C57 females that underwent AS-SSH (Figure 5). Expression of *Kiss1, Timp1, Timp2* and *Timp4* was normalized in AS-SSH BPH/5 females when compared to AS-SSH C57 (Figure 5A–D; $p > 0.05$). While *Kiss1r* is not different between NAT BPH/5 and NAT C57 eIS at e4.5, *Kiss1r* relative gene expression is lower in AS-SSH BPH-5 eIS than AS-SSH C57 during the window of embryonic implantation (Supplementary Figure S1; $p = 0.016$).

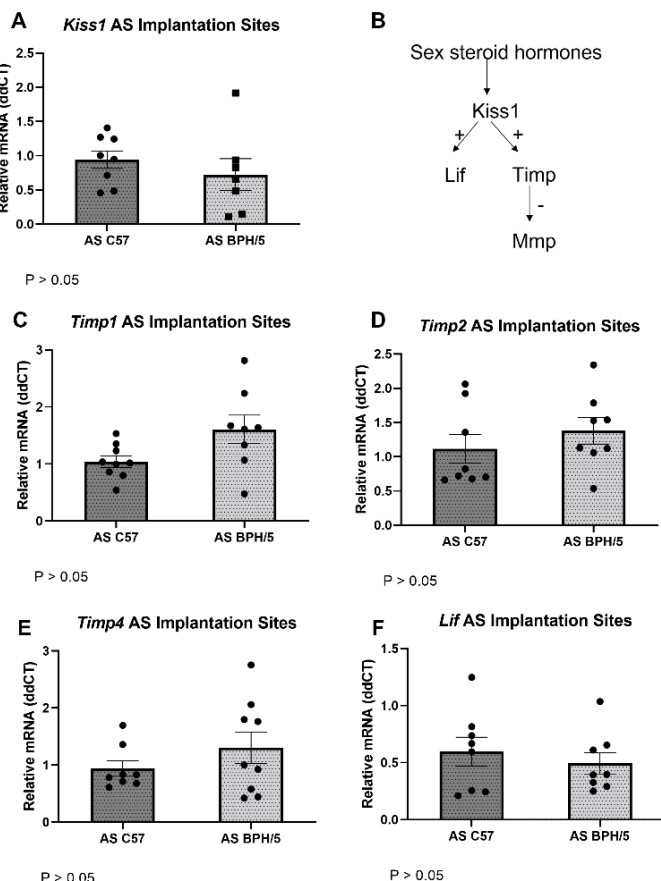

**Figure 5.** Artificial synchronization of sex steroid hormones (AS-SSH) during early gestation in the BPH/5 mouse normalizes the expression of *Kiss1* and downstream molecules. (**A**) Relative gene expression quantification of *Kiss1*. (**B**) Proposed upstream and downstream pathways of uteroplacental kisspeptin signaling. Relative gene expression quantification of the *Kiss1* downstream molecules (**C**) *Timp-1*, (**D**) *-2*, (**E**) *-4* and (**F**) *Lif* in the embryonic implantation sites of AS-SSH BPH/5 females vs. AS-SSH C57 females measured via qRT-PCR ($n$ = 8–9/group). Student's $t$-test, * $p < 0.05$ vs. time-matched C57. Data expressed as mean ± SEM.

Kisspeptin-mediated upregulation of leukemia inhibitory factor (LIF) has been proposed as one of the downstream kisspeptin signaling pathways that may affect embryonic implantation and placentation [13,36]. LIF is an E2-responsive key cytokine in embryo-uterine communication, embryo attachment and stromal decidualization [13,36]. BPH/5 females were previously reported to have lower *Lif* expression at e4.5 [27]. Therefore, we investigated whether the expression of *Lif* was also normalized in AS-SSH BPH/5 females. Notably, *Lif* was not differentially expressed between AS-SSH BPH/5 and AS-SSH C57 eIS during the window of embryonic implantation (Figure 5F; *p* > 0.05).

### 3.4. Placentation and Umbilical Cord Blood Flow Are Improved in BPH/5 Females after AS-SSH

The effects of AS of E2 and P4 in BPH/5 embryonic implantation and placentation were investigated. Importantly, AS-SSH did not have a negative effect on the number of eIS per litter, with no difference found between NAT and AS-SSH pregnancies during the window of embryonic implantation (Supplementary Figure S2; *p* > 0.05). In agreement with previous findings [25], BPH/5 females carrying NAT pregnancies presented a shallower placental expansion in the maternal decidua than NAT C57 females (Figure 6A,C,G; *p* < 0.001). Interestingly, AS-SSH not only rescued the expression of *Kiss1* and downstream molecules, but also improved placentation, with no difference seen in placental expansion between AS-SSH BPH/5 and AS-SSH C57 females (Figure 6B,D; *p* > 0.05).

Adverse fetal outcomes have been well characterized in the BPH/5 mouse model, including intrauterine fetal growth restriction and smaller litter sizes [24,25]. In this study, umbilical blood flow and fetal cardiac activity were used as indicators of fetal viability and development. Based on ultra-high frequency doppler ultrasonography, lower umbilical cord blood flow was noted on NAT BPH/5 conceptuses when compared to NAT C57 (Figure 6E; *p* = 0.0029). Concomitant with improved trophoblast invasion in AS-SSH BPH/5 pregnancies, umbilical cord blood flow was also rescued by E2 and P4 normalization in the BPH/5 mouse model, as no differences were noted in doppler ultrasound measurements between AS-SSH BPH/5 and AS-SSH C57 at e12.5 (Figure 6F; *p* > 0.05). Fetal heart rates, measured as the rate to maternal heart rate, were lower in NAT BPH/5 females than in NAT C57 at e12.5 (Figure 6G; *p* < 0.001). Despite the changes in placentation and umbilical cord blood flow, fetal heart rates were not rescued by AS-SSH, as the fetal/maternal heart rate ratio remained lower in AS-SSH BPH/5 females when compared to AS-SSH C57 (Figure 6H; *p* = 0.001).

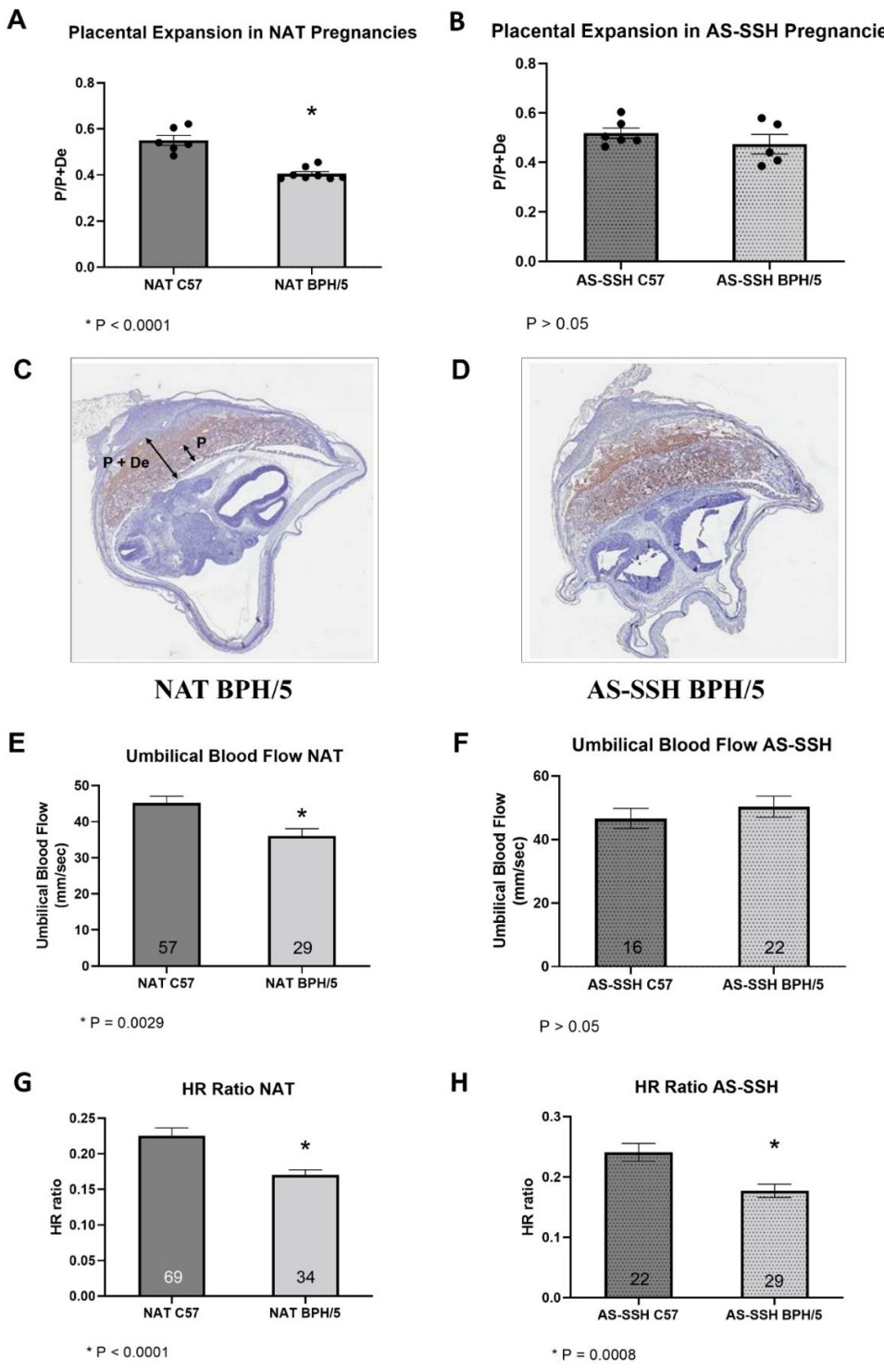

**Figure 6.** Placentation and umbilical cord blood flow are improved in BPH/5 females after artificial synchronization of sex steroid hormones (AS-SSH). (**A**,**B**) Rate of placental expansion in the maternal decidua, calculated as the ratio of placental depth (P) in relation to the placenta + maternal decidua (P + De, *n* = 5–8/group). Student's *t*-test, * *p* < 0.0001 vs. NAT C57. (**C**,**D**) Representative photomicrographs of Isolectin B4-stained fetoplacental units of (**C**) natural (NAT) and (**D**) AS-SSH BPH/5 females, 100× magnification. (**E**–**H**) Ultrasonographic measurements of umbilical blood flow and heart rate (HR) ratio, expressed as a ratio of fetal to maternal heart rate in fetoplacental units of BPH/5 and C7 females carrying NAT and AS-SSH pregnancies (*n* = 16–69/group). Student's *t*-test, * *p* < 0.05. Data expressed as mean ± SEM.

## 4. Discussion

The present study demonstrates that kisspeptins are upregulated in the uterus and maternal-fetal interface of a preeclamptic-like mouse model. Several human studies have attempted to investigate the expression and function of kisspeptins in PE [21]. However, conflicting results have been reported, which are likely associated with the lack of distinction of PE subtypes [21]. Accordingly, when considering the subclassification of PE into early- and late-onset syndromes, kisspeptins are consistently higher in term placentas of women with early-onset PE [19–21]. The innate limitations of sample collection and the lack of reliable tools for early diagnosis emphasize the importance of translational investigations to unravel a causative role of kisspeptins in the genesis of PE [37]. Among PE animal models, the BPH/5 mouse does not require pharmacological, surgical or genetic interventions for development of a PE-like syndrome [38]. Additionally, comparable to women with early-onset PE, defective placentation precedes the onset of PE-like clinical signs in BPH/5, which is characterized by poor trophoblast cell invasion, increased wall-to-lumen ratio in maternal decidual arteries, and decreased placental expansion towards the maternal decidua [24–27]. Therefore, this model provides a unique opportunity to investigate the potential role of kisspeptins in a PE-like syndrome during unexplored timepoints in human preeclamptic gestation.

Uterine kisspeptin upregulation precedes pregnancy and placentation in the BPH/5 mouse model, with higher *Kiss1* and Kp-10 expression during NP-D and early pregnancy when compared to control C57 females. Poor placental perfusion is often considered the primary stage in the pathogenesis of early-onset PE. However, abnormalities leading to this syndrome may originate earlier in gestation, or even before pregnancy [39]. While insufficient placental bed vascular remodeling is a key event in the development of PE, potential biomarkers of this syndrome have been detected in human maternal serum as early as at 7 weeks of gestation, long before the establishment of placental blood flow at gestational weeks 11–12 [39,40]. Correspondingly, in vitro fertilization (IVF) has been associated with the development of PE [41,42]. Although the contribution of coexisting risk factors for IVF demand and PE remains under debate, in vitro chorion development in IVF pregnancies, altered uterine hormone milieu and endometrial dysplasia have been proposed to play a role in PE development in this subpopulation [41–43]. Even less understood is the likelihood of pregnancy adverse outcomes in patients of fertility sparing treatments, which are often subjected to hormonal therapies including high-dose oral progestins [44–46]. Interestingly, abnormal SSH profile occurs in the BPH/5 mouse pre-pregnancy and during early gestation, and delayed embryonic implantation foregoes the development of defective placentation and PE-like clinical signs [27,28]. Based on our findings and the proposed role of kisspeptins during uterine preparation for embryonic implantation, it is speculated that SSH-mediated abnormal kisspeptin signaling may contribute to impaired embryonic implantation in this mouse model, and, potentially, in the human syndrome [13,18].

The incidence and significance of uterine abnormalities prior to the development of PE remains unknown. In the BPH/5 mouse model, non-pregnant females have higher uterine wet weights during NP-D when compared to age and estrous cycle stage-matched C57, despite lower E2 concentrations during proestrus and no differences in SSH during diestrus [28]. Studies using *Kiss1* and *Kiss1r* global knockout (KO) mice have highlighted the importance of kisspeptin signaling for adenogenesis and uterine growth, with approximately 93–98% reduced endometrial gland formation and 70–78% reduced average uterine cross-sectional area in *Kiss1/Kiss1r* KO compared to wild type littermates [47]. Moreover, although E2 administration partially rescued adenogenesis in *Kiss1/Kiss1r* KO, the rescued glands appeared to be non-functional in the absence of kisspeptin signaling, with disorganized glandular epithelium, hyperplasia and lack of glandular secretion [47]. Herein we propose that uterine *Kiss1* upregulation in the LE and GE during NP-D may contribute to the differences in the BPH/5 and C57 uterine phenotype. However, further investigations of endometrial epithelium organization and glandular development are warranted in this mouse model. In a mouse model of uterine *Kiss1r* KO, increased uterine thickness

during pregnancy was associated with E2-mediated luminal distension, followed by lack of P4-mediated luminal water resorption and closure [23]. In mouse pregnancies, uterine luminal closure coincides with blastocysts entering the uterus and is believed to facilitate embryonic implantation [23]. Interestingly, a persistent uterine lumen is repeatedly seen in NAT BPH/5 eIS at e5.5, but not in NAT e5.5 C57 eIS (unpublished data). However, the significance of this finding remains poorly understood.

The *Kiss1/Kiss1r* expression and function during normal mouse embryonic implantation and decidualization have been previously investigated [22,23]. Studies have consistently demonstrated that *Kiss1/Kiss1r* expression increases in healthy mouse embryonic implantation sites from the day of conception (e0.5) to the peak of decidualization (e7.5) [22,23]. Importantly, the lack of kisspeptin signaling specifically disrupts embryonic attachment in *Kiss1* KO mice, and only partial restoration occurs after the exogenous administration of LIF, a major cytokine in mouse embryonic implantation [36]. Based on the weak uterine Lif expression in *Kiss1* KO females, it has been suggested that kisspeptins may act as positive regulators of Lif, with an E2-Kisspeptin-LIF pathway proposed to regulate E2-mediated implantation and decidualization [13,36]. In the BPH/5 mouse model, the lower circulating E2 during early gestation may explain the lower eIS Lif expression at e4.5, despite the concomitant higher expression of *Kiss1* [27]. In agreement with this hypothesis, AS of E2 in the BPH/5 window of implantation normalized both *Kiss1* and *Lif* mRNA expression (Figure 5F).

Kisspeptin expression and function in the human preeclamptic decidua and placental bed are vastly unexplored. Few studies have linked defective decidualization to the development of human PE [48,49]. Based on our findings, a role of the maternal decidua in the evCT invasive phenotype should be considered [48,49]. Endometrial decidualization is mediated by SSH and occurs in response to embryo implantation [27,34]. Defective decidualization has been reported in the BPH/5 mouse model [27]. Namely, in the morning of e4.5, uterine markers of decidualization are seen in NAT C57 pregnancies but not in NAT BPH/5 pregnancies [27]. Interestingly, reflexive increased decidual markers were noted in the BPH/5 mouse model, as evidenced by increased alkaline phosphatase activity in the afternoon of e4.5 and morning of e5.5 when compared to control mice [27]. Based on our studies, *Kiss1* and Kp-10 are upregulated in the NAT BPH/5 decidua during the peak of decidualization, at e7.5, and may inhibit the placental expansion towards the maternal decidua (Figure 3). Term placentas *Kiss1r* upregulation has also been reported in some human studies of early-onset PE [20,21]. In the BPH/5 mouse model, *Kiss1r* is not upregulated during NP-D and embryonic implantation but is two-fold higher in NAT BPH/5 eIS at e7.5 (Supplementary Figure S1). It remains to be further elucidated if *Kiss1/Kiss1r* upregulation is associated with dysregulated decidualization in this model.

Paracrine kisspeptin signaling between distinct subpopulations of human trophoblast cells has been proposed [13,14]. Signaling between trophoblast cells and the maternal uterus, however, remains poorly described. Herein, Kp-10 immunostaining is not only located in GE and IIS LE, but also in the decidualized stromal cells of BPH/5 and C57 eIS at e7.5, suggestive of an important maternal component in kisspeptin signaling. According to studies using pseudopregnant mice, and after the artificial induction of decidualization, the presence of conceptuses does not seem to be necessary for the gradual increase in decidual *Kiss1/Kiss1r* expression [22,23]. Herreboudt et al. suggested that kisspeptin signaling may not be required for adequate placentation in mice, based on targeted disruptions of *Kiss1* or *Kiss1r* [50]. However, as alluded to by the authors, the study only included pregnancies of heterozygous females, which precluded investigations of maternal kisspeptin signaling contribution towards adequate mouse placentation [50]. Furthermore, this finding does not negate the possibility that abnormally high kisspeptin signaling may lead to defective placentation. A significant role of the maternal uterine environment in the compromised BPH/5 embryo-uterine crosstalk has been previously demonstrated by inter-strain embryo transfer studies involving BPH-5 and C57 [27]. Although kisspeptin upregulation precedes

pregnancy in the BPH/5 mouse model, the maternal and fetal contributions to kisspeptin dysregulation during early gestation should be further clarified.

Cell type-specific kisspeptin-mediated signaling pathways have been described [13,21]. The classical cellular response to kisspeptin receptor activation appears to involve the phospholipase C- protein kinase C-ERK1/2 pathway [13]. Downstream, kisspeptins have been demonstrated to inhibit MMPs and induce TIMPs gene expression [14–17]. Even though mechanistic studies are lacking, uteroplacental *Kiss1* upregulation in BPH/5 females was accompanied by upregulation of *Timp1* and *Timp2* during NP-D, e4.5 and e7.5, as well as *Timp4* at e4.5. Studies of human placental TIMP expression are scarce. However, high circulating levels of TIMP-1 and TIMP-2 have been reported in PE [35]. Importantly, although TIMP activity is usually linked to the inhibition of MMPs, systemic functions may also be performed by these molecules, as exemplified by the proposed contribution of TIMPs to endothelial cell dysfunction in cardiovascular disease [35].

Similar to humans, mice have a hemochorial placentation and present significant remodeling of the decidual spiral arteries [51]. However, mouse trophoblast cell invasion is shallower, primarily interstitial, and occurs later in gestation when compared to humans [52]. Nonetheless, striking similarities exist between the two species, including a highly conserved network of SSH-regulated genes that modulate decidualization [34]. Similarly, the SSH profile during early gestation appears to play a major role in the BPH/5 gene expression and phenotype. The normalization of E2 and P4 concentrations during early pregnancy not only mitigated *Kiss1, Timp1, Timp2, Timp4* and *Lif* expression dysregulation in comparison to C57 pregnancies, but also ameliorated placentation and umbilical cord blood flow (Figure 6). Nonetheless, further investigation of trophoblast cell density and decidual vessel remodeling in AS-SSH BPH/5 females should be pursued.

## 5. Conclusions

In summary, our investigations suggest that uterine kisspeptin upregulation may precede the occurrence of pregnancy and persist during early gestational milestones in a PE-like syndrome. Furthermore, this work provides further evidence of uteroplacental *Timp* and *Lif* dysregulation in a PE-like syndrome, which are potential molecules downstream to kisspeptins. Consistent with previous findings, E2 and P4 appear to mediate uteroplacental kisspeptin expression. In this study, normalization of circulating concentrations of SSH in pregnant BPH/5 females not only mitigated *Kiss1* upregulation, but also rescued the expression of multiple molecules previously shown to be downstream to kisspeptin and ameliorated fetoplacental outcomes.

**Supplementary Materials:** The following supporting information can be downloaded at: https://www.mdpi.com/article/10.3390/reprodmed3040021/s1, Supporting information, including Mus musculus-specific primer sequences used, kisspeptin receptor gene expression in the BPH/5 non-pregnant diestrus, and embryonic implantation rate after artificial synchronization of sex steroid hormones (AS-SSH). References [22,27,53] are cited in the supplementary materials.

**Author Contributions:** Conceptualization, V.C.L.G., A.K.W. and J.L.S.; methodology, V.C.L.G., A.K.W. and J.L.S.; validation, V.C.L.G., A.K.W. and B.M.G.; formal analysis, V.C.L.G., A.K.W., K.R.C., C.A.L., K.F.B. and C.-C.L.; investigation, V.C.L.G., A.K.W., K.R.C., C.A.L. and L.R.F.; resources, A.K.W., E.L.O. and J.L.S.; data curation, V.C.L.G.; writing—original draft preparation, V.C.L.G.; writing—review and editing C.-C.L., E.L.O. and J.L.S.; visualization, V.C.L.G., A.K.W., C.-C.L. and J.L.S.; supervision, J.L.S.; project administration, J.L.S.; funding acquisition, V.C.L.G., A.K.W. and J.L.S. All authors have read and agreed to the published version of the manuscript.

**Funding:** This research was funded by the Louisiana State University Veterinary Clinical Sciences CORP Grant (2020-2021), and the National Institutes of Health (NIH-P20GM135002).

**Institutional Review Board Statement:** The animal study protocol was approved by the Institutional Animal Care and Use Committee at Cornell University (2009-0028) and Louisiana State University Review Board (16-106, 18-112, 21-143) for studies involving animals.

**Informed Consent Statement:** Not applicable.

**Data Availability Statement:** Data are contained within the article or Supplementary Materials.

**Acknowledgments:** The authors acknowledge Robin L. Davisson for the generous gift of BPH/5 mice.

**Conflicts of Interest:** The authors declare that they have no conflict of interest.

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
