# Peer review of "Kisspeptin Is Upregulated at the Maternal-Fetal Interface of the Preeclamptic-like BPH/5 Mouse and Normalized after Synchronization of Sex Steroid Hormones"

_2673-3897, doi:10.3390/reprodmed3040021_

Round 1

Reviewer 1 Report

Authors of the present manuscript studied the expression pattern of kisspeptin and defined its association with Sex steroid hormones and downstream molecules such as Lif and Timp which play significant roles in endometrium to aid embryo implantation. They used BPH/5 preeclamptic like mouse model and suggest kiss1 levels are elevated before pregnancy and are consistent during early gestation. 

Minor Comments: 

1) Please make a complete animal workflow by adding natural pregnancies and their sample collection time line in Fig 1. 

2)  Add p values directly on the bar graphs 

3) In Fig 3 make sure panel E is of appropriate magnification. 

4) Please check labeling in Fig 3, A-B says quantitation of Kiss 1 gene where on figure it shows A and F. 

5) 

Major Comments:

1) Fig 3 shows the expression of Kiss 1 at embryo implantation sites but it is unclear where the implantation site is on the histological section (Please mark with an arrow). Staining seems to be consistent through out the different cell types (Stromal and endothelial)

2) Either include quantitation for IHC from fig 2 & 3 or a western blot to confirm changes in the protein levels of Kiss 1.  

Author Response

Reviewer 1:

Authors of the present manuscript studied the expression pattern of kisspeptin and defined its association with Sex steroid hormones and downstream molecules such as Lif and Timp which play significant roles in endometrium to aid embryo implantation. They used BPH/5 preeclamptic like mouse model and suggest kiss1 levels are elevated before pregnancy and are consistent during early gestation.

Minor Comments:

1) Please make a complete animal workflow by adding natural pregnancies and their sample collection timeline in Fig 1.

The authors have acknowledged the deficiency in the original timeline figure and have rectified it in accordance with the suggestions from reviewer 1. The figure legend was also edited, in agreement with the graphic changes.

2)  Add p values directly on the bar graphs

The p values have been added directly under all the bar graphs.

3) In Fig 3 make sure panel E is of appropriate magnification.

All the figure panels were revised and rectified accordingly.

4) Please check labeling in Fig 3, A-B says quantitation of Kiss 1 gene where on figure it shows A and F.

The figure labels were revised and rectified.

Major Comments:

1) Fig 3 shows the expression of Kiss 1 at embryo implantation sites but it is unclear where the implantation site is on the histological section (Please mark with an arrow). Staining seems to be consistent through out the different cell types (Stromal and endothelial)

2) Either include quantitation for IHC from fig 2 & 3 or a western blot to confirm changes in the protein levels of Kiss 1. 

The authors are extremely grateful for the reviewer’s comments regarding the immunohistochemistry assay. Quantification was performed using the software ImageJ and included in the body of the text and images. Furthermore, images of greater representability were utilized, along with proper labelling.

Reviewer 2 Report

Dear authors,

I read with great interest the manuscript, which falls within the aim of this Journal. In my honest opinion, the topic is interesting enough to attract the readers’ attention. Nevertheless, authors should clarify some points and improve the discussion, as suggested below. Authors should consider the following recommendations:

I suggest you to improve with a paragraph the paper on fertility preservation.

Usually in case of early diagnosis of malignancies a fertility sparing treatment is needed in these pts and is hard suggested them to preserve, before surgical treatments, their fertility by an ovarian stimulation by antagonist protocol and frezee  all their gamets trought vitrification system for future pregnancy.

I suggest you to read and cite these papers:

Closed vs. Open Oocyte Vitrification Methods Are Equally Effective for Blastocyst Embryo Transfers: Prospective Study from a Sibling Oocyte Donation Program

Fertility-sparing approach in women affected by stage i and low-grade endometrial carcinoma: An updated overview.

Fertility Sparing Treatments in Endometrial Cancer Patients: The Potential Role of the New Molecular Classification

Biomolecular and Genetic Prognostic Factors That Can Facilitate Fertility-Sparing Treatment (FST) Decision Making in Early Stage Endometrial Cancer (ES-EC): A Systematic Review

GnRH antagonist administered twice the day before hCG trigger combined with a step-down protocol may prevent OHSS in IVF/ICSI antagonist cycles at risk for OHSS without affecting the reproductive outcomes: a prospective randomized control trial

I also suggest you to refer in the text how the embryo implantation could be improve with other tecniques as embyo culture supernatant and it could be interesting to focus the kisspeptins role in the preeclampsia paogenesis.

I suggest you to read and cite these papers too:

 Injection of embryo culture supernatant to the endometrial cavity does not affect outcomes in IVF/ICSI or oocyte donation cycles: A randomized clinical trial

The role of endoglin and its soluble form in pathogenesis of preeclampsia

Author Response

The authors are grateful for the reviewer’s comments and suggested supplementary literature. The papers suggested were carefully read and incorporated into the manuscript accordingly.

Round 2

Reviewer 1 Report

I would like to thank authors for addressing the comments and making the manuscript more stronger. 

Reviewer 2 Report

now the paper is ok